# Transcriptome Analysis Revealed the Early Heat Stress Response in the Brain of Chinese Tongue Sole (*Cynoglossus semilaevis*)

**DOI:** 10.3390/ani14010084

**Published:** 2023-12-26

**Authors:** Yue Wang, Chengcheng Su, Qian Liu, Xiancai Hao, Shenglei Han, Lucas B. Doretto, Ivana F. Rosa, Yanjing Yang, Changwei Shao, Qian Wang

**Affiliations:** 1Tianjin Key Laboratory of Aqua-Ecology and Aquaculture, Fisheries College, Tianjin Agricultural University, Tianjin 300384, China; lunawang16@163.com (Y.W.); eyeforever@126.com (Y.Y.); 2National Key Laboratory of Mariculture Biobreeding and Sustainable Goods, Yellow Sea Fisheries Research Institute, Chinese Academy of Fishery Sciences, Qingdao 266071, China; 13335088169@163.com (C.S.); liuqian97927@163.com (Q.L.); best_hxc@163.com (X.H.); 17860712133@163.com (S.H.); lucas.doretto@unesp.br (L.B.D.); shaocw@ysfri.ac.cn (C.S.); 3Department of Structural and Functional Biology, Institute of Biosciences, São Paulo State University (UNESP), Botucatu 01049-010, Brazil; ivana.felipe@unesp.br; 4Laboratory for Marine Fisheries Science and Food Production Processes, Laoshan Laboratory, Qingdao 266237, China

**Keywords:** *Cynoglossus semilaevis*, transcriptome, heat shock, gene

## Abstract

**Simple Summary:**

Temperature is a pervasive environmental factor that plays a crucial role in the lives of aquatic organisms. However, the mechanisms governing the response to temperature remain unclear in teleost fish. In this research, the marine benthic fish Chinese tongue sole (*Cynoglossus semilaevis*), which is economically significant in northern China, is used as a model to analyze this issue. The brains of female and male *C. semilaevis* were examined via transcriptomics technology to investigate their response to temperature. Our findings revealed the existence of specific genes and pathways involved in cortisol synthesis and secretion, neuroactive ligand–receptor interactions, TGF-beta signaling pathway, and JAK/STAT signaling pathway. The genes identified includes the HSP family, *tshr*, *c-fos*, *c-jun*, *cxcr4*, *camk2b*, and *igf2*. Our study provides some evidence that *C. semilaevis* responds to temperature, highlighting a potential path for further investigation.

**Abstract:**

As a common influencing factor in the environment, temperature greatly influences the fish that live in the water all their life. The essential economic fish Chinese tongue sole (*Cynoglossus semilaevis*), a benthic fish, will experience both physiological and behavioral changes due to increases in temperature. The brain, as the central hub of fish and a crucial regulatory organ, is particularly sensitive to temperature changes and will be affected. However, previous research has mainly concentrated on the impact of temperature on the gonads of *C. semilaevis*. Instead, our study examines the brain using transcriptomics to investigate specific genes and pathways that can quickly respond to temperature changes. The fish were subjected to various periods of heat stress (1 h, 2 h, 3 h, and 5 h) before extracting the brain for transcriptome analysis. After conducting transcriptomic analyses, we identified distinct genes and pathways in males and females. The pathways were mainly related to cortisol synthesis and secretion, neuroactive ligand–receptor interactions, TGF beta signaling pathway, and JAK/STAT signaling pathway, while the genes included the HSP family, *tshr*, *c-fos*, *c-jun*, *cxcr4*, *camk2b*, and *igf2*. Our study offers valuable insights into the regulation mechanisms of the brain’s response to temperature stress.

## 1. Introduction

The stress response, which is the physiological response to stressors, comprises a complex sequence of biochemical, physiological, and behavioral adaptations to environmental changes that facilitate the preservation of internal homeostasis and survival under certain conditions [1,2,3]. Stress responses can also have adverse long-term effects, depending on the intensity, duration, and frequency of stressors [4,5]. Various stressors, including water temperature, dissolved oxygen, pH, light intensity, salinity, and water speed, can cause stress. Among them, temperature is the most important and is recognized as the biotic master factor for fish [6]. Abrupt temperature fluctuations can have detrimental effects on fish growth [7,8], behavior [9,10], physiology, biochemistry [11,12,13], and reproduction [6,14,15], potentially leading to illnesses or fatalities [16,17]. Benthic creatures, which spend their entire lives submerged, are among the most severely affected aquatic organisms, particularly fish. Although fish can adapt to seasonal temperature changes, sudden fluctuations or temperature changes outside their normal range can overwhelm them and cause negative effects. In severe cases, these can lead to disease or even death. Therefore, examining the impact of temperature fluctuations on fish can assist in regulating their population and density, resulting in improved economic efficacy.

The brain is acknowledged as the hub of neuromodulation, and the central nervous system regulates most of the body’s physiological activity. It is suggested that the pineal gland in the brain is responsible for perceiving light and temperature as a photoreceptive nerve structure [6,18]. Additionally, the brain produces a range of hormones to modulate the body’s physiological requirements. Previous research has demonstrated that the brain of fish can activate regulatory mechanisms to regulate downstream genes and maintain the body’s internal homeostasis when external temperatures change [6]. For instance, Antarctic fish (*Harpagifer antarcticus*) exposed to 11 °C showed higher brain neurotransmitter levels than those exposed to 2 °C [19]. Likewise, increased calcium activity was found in zebrafish brains under a high temperature of 28 °C [20]. In addition, the maintenance of homeostasis was influenced by hormones, which can interact with receptors on the cell membrane, facilitating the production of second messengers or binding directly to cytoplasmic receptors [21]. In mammals, for instance, chronic heat stress has been proven to interfere with sex hormone and neuroendocrine hormone levels in female rats [22]. Moreover, high temperatures inhibited brain *sbGnRH*, pituitary *GnRH-R*, and *LHβ* gene expressions in red seabream (*Pagrus major*) [23] while increasing aromatase activity in the brains of sea bass (*Dicentrarchus labrax*) brain [24]. Similarly, Tsai et al. [25] discovered that heat stress upregulated *Erβ* (estrogen receptors) mRNA expression in the tilapia brain, and *GnRH3*, *PACAP*, *IGF-1*, *βLH*, and *prolactin* were inhibited by high or low water temperatures in the blue gourami (*Trichogaster trichopterus*) [26].

Furthermore, some genes respond to changes in temperature by exhibiting differential expression in the brain. The identification of the HSP family as heat shock chaperones has recently been the subject of extensive research. Most genes belonging to the HSP family showed rapid upregulation in response to high temperatures. Ahmet Topal et al. [27] showed that increasing water temperatures increased the upregulation of *hsp70* and *hsp90* gene expression, decreased the expression of antioxidant enzymes, and led to endoplasmic reticulum (ER) stress in the brains of *Oncorhynchus mykiss*. These results emphasized the significance of researching the impacts of temperature stress on fish brain regulation.

The Chinese tongue sole (*Cynoglossus semilaevis*) is an economically important marine fish in northern China and its genome has been sequenced [28]. Nevertheless, the studies conducted until now have predominantly concentrated on assessing the influence of heat stress on gonadal functions. In this case, investigating the molecular mechanisms underlying heat stress in *C. semilaevis* brain is necessary. Hence, our investigation focused on the brain of *C. semilaevis*.

In our study, we investigate the rapid heat response of the brain in *C. semilaevis* through transcriptome sequencing. Our analysis includes temporal, KEGG, and protein network interaction analyses and heatmap analyses. We anticipate identifying valuable genes and pathways in the transcriptome of the heat-stressed brain of *C. semilaevis*. Furthermore, we aim to explore the genes and pathways in the brain of *C. semilaevis* that respond quickly to temperature and offer new avenues for research into heat stress responses.

## 2. Materials and Methods

### 2.1. Sample Stocks and High-Temperature Treatment

Six-month-old *C. semilaevis* (*n* = 50) were obtained from Laizhou Mingbo Aquatic Co., Ltd. in Yantai, China. These fish were reared in 120 L tanks at the Yellow Sea Fisheries Research Institute and fed with commercial pellets twice a day. After two days of acclimatization in filtered seawater (temperature: 22 °C, salinity: 25–29 ppt.), the individuals were randomly divided equally into two groups: a control (CT) group exposed to 22 °C and heat stress (HS) groups exposed to 28 °C for 1 h, 2 h, 3 h, and 5 h, respectively. After treatment, 5 female and 5 male individuals of each group were randomly sampled. After anesthesia with 0.05% MS-222 (Sigma, Shanghai, China) via a water bath [29], the brains were promptly extracted from each fish for subsequent RNA extraction and their caudal fins were preserved for sex identification (Figure 1A). Sex-specific primers were used to determine genetic sex as previously detailed by Liu et al. (2014) [30]. All animal experiments were approved by the Institutional Animal Care and Use Committee (IACUC) of YSFRI, CAFS (Approval No.: YSFRI-2023018).

### 2.2. Extraction of Total RNA and RNA-seq Preparation

Total RNA was extracted using Trizol reagent (Invitrogen, Waltham, MA, USA) from each brain sample. The quality of the RNA samples was evaluated using an Agilent 2100 bioanalyzer (Thermo Fisher Scientific, Santa Clara, CA, USA). Subsequently, mRNA library construction was conducted using high-quality RNA (RIN > 8) following the conventional protocol. Briefly, eukaryotic mRNA was enriched using magnetic beads coated with Oligo(dT). The mRNA was subsequently fragmented by adding a fragmentation buffer. The first cDNA strand was synthesized from mRNA using random hexamers as templates, and the second cDNA strand was generated by the addition of buffers, dNTPs, RNase H, and DNA polymerase I. The second cDNA strand was purified using a QiaQuick PCR kit and eluted with EB buffer and end-repaired and ligated to the sequencing junction; then, fragment size selection was performed by agarose gel electrophoresis and, finally, cDNA was enriched by PCR amplification. The mRNA library was prepared. For biological analysis and experimental preciseness, three replicates of each treatment were prepared, resulting in a total of thirty samples: CT—F1–F3, CT—M1–M3, HS1h—F1–F3, HS1h—M1–M3, HS2h—F1–F3, HS2h—M1–M3, HS3h—F1–F3, HS3h—M1–M3, HS5h—F1–F3, and HS5h—M1–M3 (F and M represent female and male, respectively). Following that, the libraries of all groups were sequenced on the Agilent 2100 platform by Berry Genomics Co., Ltd. in Beijing, China.

### 2.3. Data Quality Control and Gene Annotation

The sequencing reads were filtered via default parameters using SOAPnuke v1.4.0 [31] and Trimmomatic v0.36 [32] software to exclude adapters, reads with more than 3% unknown nucleotides, and reads with over 20% low-quality bases (Q value <= 5). The resulting clean reads were mapped to the *C. semilaevis* genome (NCBI Cse_v1.0) using HISAT2 v2.1.0 [33] and Bowtie2 v2.2.5 [34], which coincided with the reference transcript sequence to eliminate rRNA sequences and annotate genes.

### 2.4. Differentially Expressed Gene Analysis

Transcript numbers were rationed using the RNA-Seq by expectation maximization (RSEM v1.2.8) method [35], and the expression levels of genes were standardized using the fragments per kilobase million (FPKM) technique. This normalization approach ensured that gene expression was not affected by differences in gene lengths or sequencing data volume. Differential expression analysis was conducted using DESeq2 v1.42.0 [36]. Compared with the control (CT) group, transcripts showing a |fold change| ≥ 2 and an adjusted *p*-value (q value) < 0.05 were identified as significantly differentially expressed genes (DEGs).

### 2.5. Time-Series Expression Clustering

By using the R package Mfuzz (v 2.52; Options = −c 4, −m 1.25) [37], total DEGs with comparable expression patterns were clustered in both males and female brains. The resulting gene clusters were then annotated and enriched using the Kyoto Encyclopedia of Genes and Genomes (KEGG) database. We identified four distinct clusters for both males and females. Annotated DEGs were used to perform KEGG (http://www.kegg.jp/, accessed on 1 July 2023) enrichment analyses based on the hypergeometric test by Phyper (http://en.wikipedia.org/wiki/Hypergeometric_distribution, accessed on 1 July 2023). The threshold of *p*-value < 0.05 is considered to be significant.

### 2.6. Prediction of Protein–Protein Interaction Network (PPI) and Heatmap Construction

The top 12 pathway-enriched genes in male and female cluster 3 were used and analyzed by KEGG for PPI analysis. To further investigate the relationship of genes, PPI networks were constructed with default parameters in *C. semilaevis* after heat shock by using STRING v10.0 (http://string-db.org/, accessed on 18 July 2023) [38,39]. Subsequently, we used Cytoscape v3.9.1 software to visualize the relationships between the genes more intuitively. To ensure a comprehensive examination of key genes, we excluded marginal genes from the analysis and sorted the genes with degree values. Based on the ranked list of degree values, we selected the top 60 genes for heat mapping. The overall experimental flow was shown in Figure 1B. The list of genes sorted by degree value was shown in Appendix A.

### 2.7. Validation of qRT-PCR

To verify the reliability of the data, we selected nine DEGs related to high-temperature-induced heat stress for qRT-PCR validation randomly. The *β-actin* gene was used as the internal control [40]. Primers for the nine genes needed were designed based on sequences from the National Center for Biotechnology Information (NCBI) database (Appendix A). All reactions were performed following the previously reported protocol [41]. Briefly, one microgram of total RNA for high-throughput transcriptome sequencing was reverse-transcribed into cDNA with the PrimeScript™ RT reagent Kit with gDNA Eraser (Takara, Kusatsu, Japan). Then, qRT-PCR was performed using QuantiNova™ SYBR Green PCR Kit (Qiagen, Hilden, Germany) in 20 μL reactions containing 10 μL of 2 × SYBR Green PCR Master Mix, 2 μL of QN ROX Reference Dye, 0.7 μM of forward primer, 0.7 μM of reverse primer, and 1 μL of cDNA. The cycling program was carried out at 95 °C for 2 min, followed by 40 cycles of 95 °C for 5 s and 60 °C for 10 s; a melting curve analysis in an ABI StepOnePlus Real-Time PCR system (Applied Biosystems, Waltham, MA, USA). Reactions were performed in triplicate. The relative expression fold changes of these genes were analyzed using the 2^−ΔΔCt^ method.

### 2.8. Statistical Analysis of qRT-PCR

The results of the qRT-PCR analysis were expressed as the mean ± SEM (standard error of the mean). By using GraphPad Prism 6. 0 (GraphPad Software, Inc., San Diego, CA, USA), one-way ANOVA was performed and the values were compared. *p*-values < 0.05 were considered statistically significant (* *p* < 0.05, ** *p* < 0.01, *** *p* < 0.001).

## 3. Results

### 3.1. Raw Sequencing Data Quality Control

To investigate the influence of heat stress in the brain of *C. semilaevis*, transcriptomic sequencing of control (CT—22 °C) and heat stress (HS—28 °C) groups was performed. In the current study, 819.68 million clean reads were obtained from thirty libraries and the percentage of Q30 bases was not less than 88.99% (Appendix A). Moreover, the clean reads of each sample compared with the sequences of the reference genome (NCBI Cse_v1.0) (Appendix A) showed that the matches ranged from 89.79% to 90.84%, indicating a high confidence level for the transcriptome sequencing results. Among these reads, the majority were found in the genomic CDS region, followed by the 3’UTR area (Appendix A). Additionally, the principal component analysis (PCA) depicted a clear cluster separation between the CT and HS groups (Appendix A). Accordingly, the most significant difference between females and males was observed after 1 and 2 h of heat-stress treatment. The transcriptome data with accession number PRJNA1020292 have been uploaded to NCBI.

### 3.2. Differentially Expressed Genes under Heat Stress

In the context of acute heat stress, we assessed the DEGs among the control and heat stress (1 h, 2 h, 3 h, and 5 h) groups (Figure 2; Appendix A). In females (Figure 2A,B), initially, 2761 genes displayed differential expression between the control group and 1 h high-temperature group; subsequently, following 2 h and 3 h, 3709 and 5724 genes, showed differential expression, respectively. A comparable trend was noted among males subject to heat stress (Figure 2A,C)—2935 genes were differentially expressed during the initial hour, succeeded by 5677 and 5092 after 2h and 3h of exposure, respectively. The largest quantity of DEGs was detected in females after 3 h of heat stress treatment, while in males, the highest number was detected after 2 h. A significant decrease in the DEG number occurred in both sexes over time, with females exhibiting 1625 DEGs and males exhibiting 1944 DEGs after 5 h of treatment. Furthermore, to identify shared DEGs among comparison groups, a Venn diagram was produced using different DEG datasets (Figure 2B,C). There were 570 DEGs shared among the four groups (1 h, 2 h, 3 h, and 5 h) compared with the CT group in females and 724 DEGs in males.

### 3.3. Time-Series Expression and KEGG Enrichment of DEGs in C. semilaevis under Heat Stress

To conduct the time-series analysis, all heat stress groups were considered including females and males. Subsequently, a total of 9510 DEGs were submitted to temporal analysis, leading to the identification of four distinct expression pattern clusters in each sex (Figure 3A,B). In females (Figure 3A), the time-series analysis revealed that cluster 1 (2339 genes) and cluster 2 (2123 genes) exhibited a U-shaped pattern in comparison with the control group. Conversely, clusters 3 (2045 genes) and 4 (3003 genes) exhibited a bell-shaped pattern. Furthermore, the number of upregulated DEGs peaked after 2 h of heat stress treatment in cluster 3 and peaked after 3 h in cluster 4, respectively. In males, heat stress caused a U-shaped pattern in cluster 1 (2190 genes) and cluster 2 (2024 genes), whereas it caused a bell-shaped pattern in cluster 3 (2758 genes) and cluster 4 (2536 genes). Moreover, cluster 3 reached its peak of DEGs after 1 h, while cluster 4 reached its peak after 3 h of heat stress (Figure 3B). Intriguingly, the expression levels of genes situated in clusters 1, 3, and 4 in females and males returned to baseline (0 h) after 5 h of heat stress treatment.

To further investigate the biological functions of DEGs subjected to heat stress, we performed KEGG enrichment analysis (Figure 3C–F, Appendix A). This involved merging clusters 1 and 2, which represented “the U-shaped” DEGs, and clusters 3 and 4, which corresponded to “the bell-shaped” DEGs in both sexes. In U-shaped DEGs, the pathways most enriched were spliceosome, DNA replication and proteasome (Figure 3C,D). We also find pathways including glycosylphosphatidylinositol (GPI) anchor biosynthesis, cytokine–cytokine receptor interaction, the notch signaling pathway, fatty acid biosynthesis, and fat digestion and absorption in females (Figure 3C), and metabolic pathways, tyrosine metabolism, RNA degradation, GPI anchor biosynthesis, and the notch signaling pathway in males (Figure 3D). Within the bell-shaped clusters of females and males, neuroactive ligand–receptor interaction, cytokine–cytokine receptor interaction, cortisol synthesis and secretion, aldosterone synthesis and secretion, TGF-beta signaling pathway, calcium signaling pathway, JAK/STAT signaling pathway, IL-17 signaling pathway and cAMP signaling pathway were enriched in the combinatorial cluster in both sexes (Figure 3E,F). Detailed tables for KEGG enrichment analysis can be found in Appendix A.

### 3.4. Protein-Protein Interaction Analyses (PPI)

The HSP family is recognized for its rapid response to heat stress, with most of its genes being upregulated in response to temperature [42,43]. This family of chaperone proteins plays a crucial role in safeguarding an organism from stress and restoring cellular homeostasis [44,45]. We aimed to screen genes and pathways from our data that were upregulated, similar to the HSP family, in response to high temperatures in a short period. This will provide references and evidence for our future experiments and facilitate our subsequent studies. Therefore, we selected cluster 3, which rapidly responded to high temperatures, for subsequent analyses in both females and males (Figure 4A,B; Appendix A). From Figure 4A,B, it was clear that the JAK/STAT signaling pathway, neuroactive ligand–receptor interaction played an important role in the regulation of neuron function through modulating transcription factors and gene expression [46], and the P13-AKT signaling pathway and cytokine–cytokine receptor interaction were enriched in both females and males. The IL-17 signaling pathway and estrogen signaling pathway were particularly enriched in females, while the MAPK signaling pathway was particularly enriched in males. We also performed KEGG enrichment analysis and PPI analyses for other clusters (Appendix A).

To further examine the regulatory correlation of heat stress in the brain, we conducted a PPI network based on the top 12 KEGG-enriched pathways of female and male cluster 3. We chose pathway-enriched genes and analyzed PPI, with a total of 287 genes selected for PPI analysis in females and 352 genes in males (Appendix A). After excluding marginal genes, Cytoscape software v3.9.1 produced Figure 4C,D and developed a list of genes ranked by degree value (Appendix A). Therefore, we found some interesting genes related to protein network interactions, including *jun* (Jun proto-oncogene, AP-1 transcription factor subunit), *cxcr4* (chemokine (C-X-C motif), receptor 4b), *hspb1* (heat shock protein, alpha-crystallin-related, 1), *hspa4l* (heat shock protein 4 like), *cacng4* (calcium voltage-gated channel auxiliary subunit gamma 4), *fosab* (v-fos FBJ murine osteosarcoma viral oncogene homolog Ab), *cacng3* (calcium channel, voltage-dependent, gamma subunit 3b), *camk2b* (calcium/calmodulin-dependent protein kinase II beta), LOC103379237 (gonadotropin-releasing hormone II receptor-like), *camkk1* (calcium/calmodulin-dependent protein kinase kinase 1, alpha a), and *igf2b* (insulin-like growth factor 2b).

To provide a clearer overview of the protein network interaction genes in the brain (Appendix A), we identified the top 60 genes based on their degree value and produced a heatmap. The heatmap of the DEGs provided a visual illustration of the expression differences between the control and heat stress groups. Our results showed that heat stress up-regulated genes involved in heat shock, steroidogenesis, thyroid hormones, and inflammatory pathways in females compared with the control group during the HS 3 h. More precisely, among the members of the HSP family A, *hspa4l* (heat shock protein family A member 4-like), *hspa5* (heat shock protein family A member 5), and members of the HSP family B, *hspb6* (heat shock protein family B member 6) were up-regulated after 1 h to 3 h of heat stress. Similarly, *dnajb1* and *dnajb12* belonging to the DnaJ heat shock protein family (Hsp40) were also significantly upregulated under HS. Interestingly, heat stress also increased the expression of thyroid-stimulating hormone receptor (*tshr*) and the female-related genes involved in progestin syntheses, such as progesterone receptor (*pgr*), progesterone receptor membrane component 1 (*pgrmc1*), and progesterone receptor membrane component 2 (*pgrmc2*). In addition, the expression levels of a series of inflammatory factors, including *cxcr4* and *socs3*, were distinctly upregulated during heat stress treatment for 2 h and 3 h in females. At 5 h of heat stress, the levels of gene expression for the other genes were comparable to those of the controls and exhibited a bell-shaped expression trend, except for *pgrmc2* and *dnajb12*, which experienced up-regulation after 5 h of heat stress (Figure 5A).

In males, our results showed that heat treatment up-regulated the transcription levels of important growth factors during the HS 1 h to 3 h, including *igf2* (insulin-like growth factor 2), *igf2r* (insulin-like growth factor 2 receptor), and *igf2bp2* (insulin-like growth factor 2 mRNA binding protein 2). Further, genes involved in the immune and inflammatory system were also upregulated after heat stress treatment in males, including *socs3* (suppressor of cytokine signaling 3-like), *cxcr4* (chemokine C-X-C motif), *ccr7* (C-C motif chemokine receptor 7), and *jak2* (tyrosine-protein kinase JAK2). All of the genes exhibited a bell-shaped expression trend. At 5 h of heat stress, the levels of gene expression were comparable to those of the controls (Figure 5B).

### 3.5. qRT-PCR Validation

Nine DEGs were selected to validate the transcriptome results randomly. As shown in Appendix A, the expression patterns of DEGs identified via RNA-Seq analyses were generally similar to those obtained in qRT-PCR. These results support the reliability of our transcriptome data.

## 4. Discussion

### 4.1. The Relationship between Heat Stress Time and Number of DEGs

The brain serves as a central regulatory hub within an organism, orchestrating downstream gene activities to maintain internal homeostasis in response, for example, to fluctuations in external temperatures [1,2,3]. Meanwhile, the specific mechanisms of brain modulation under heat stress remain less understood. Therefore, in this study, we conducted a transcriptomic analysis to examine variations in the brain of *C. semilaevis* during heat stress (1 h, 2 h, 3 h, and 5 h) to identify key genes and pathways influenced by elevated temperatures. Based on the analyses of differentially expressed genes, we observed sex-specific effects in response to heat stress in *C. semilaevis*. Notably, in males, the HS1h vs. CT comparison exhibited a higher number of overall differentially expressed genes (DEGs) than in females. Furthermore, the highest number of DEGs in males was observed after 2 h of heat stress treatment, while in females, this peak was observed after 3 h. The number of DEGs in 5 h was the lowest in both males and females.

The temporal analysis revealed that 1 h of heat stress treatment was sufficient to induce significant changes in gene expression in both sexes. By the 5 h of heat stress, some gene expression patterns resembled those of the control group (Figure 3A,B). These findings indicated that even a few hours of heat stress can profoundly impact the brain homeostasis of *C. semilaevis*. Given the central role of this vital organ in regulating physiological processes and responding to environmental challenges [47], it is plausible that the hypothalamic–pituitary axis played a pivotal role in restoring brain homeostasis in response to the abrupt temperature increase.

Based on KEGG analysis of female and male cluster 3, it is evident that the pathways enriched in females and males were different. Specifically, males were enriched for the MAPK signaling pathway; meanwhile, females were enriched for the IL–17 signaling pathway and estrogen signaling pathway. Moreover, the pathways enriched in males included hormone-related signaling pathways, whereas females had fewer. On the other hand, both males and females were enriched for the cytokine–cytokine receptor interaction and Pl3K-Akt signaling pathway. Both the male and female lists included both shared and distinct genes.

### 4.2. Heat Stress Upregulated Immune-Related Genes in Both Sexes

The list of genes we obtained through the PPI analysis reflected several immune-related genes that can be upregulated under heat stress in both sexes (Appendix A). Among them, *jun* (*c-jun*) and *c-fos* were genes that appeared on both male and female lists. For measuring neuronal brain activity, *c-fos* (a transcription factor) was commonly used as an indicator [27]. *c-fos* was a member of the Fos family. Various stress factors have been shown to increase *c-fos* expression [48,49]. As early as 1990, it was demonstrated that *jun* and *c-fos* exhibited elevated mRNA levels in response to heat shock in mammals [50]. It has been demonstrated that *fos* (*c-fos*) accumulates in the anterior hypothalamus of rats after heat stress [51]. Furthermore, it is known that dimers of *c-fos* and *c-jun* were activated in response to immune stimuli [52]. We hypothesized that the expression of *c-fos* and *jun* increased in males and females during early heat stress may be an immune function that protects the organism from damage caused by temperature stress. Our finding that *c-fos* is temperature-sensitive was consistent with previous studies of both Zhao [53] and Ahmet Topal [27].

*CXC* (α-chemokine) and *CC* (β-chemokine) were the two major subfamilies of chemokines and play important regulatory role in the immune defense of fish [54,55,56,57,58]. In the present study, a series of genes involved in immune and inflammatory pathways were modulated under high-temperature conditions. For instance, the chemokines (*cxcr4* and *ccr7*) and the suppressor of cytokine signaling 3-like (*socs3*) were significantly upregulated during heat stress for 1–3 h. In recent years, *cxcr4*, *ccr7*, and *socs3* have been extensively characterized due to their innate immune role in a few fish species, including blunt snout bream (*Megalobrama amblycephala*) [59], rock bream (*Oplegnathus fasciatus*) [60], and grass carp (*Ctenopharyngodon idella*) [61]. Among them, only *cxcr4* and *socs3* were associated with stress responses by confinement in common carp (*Cyprinus carpio*) [62] and by high-temperature treatment in olive flounder (*Paralichthys olivaceus*) [63]. In this sense, our study was the first to report that *cxcr4* is a temperature-responsive gene *C. semilaevis*, and this finding was validated through qRT-PCR. Accordingly, the *cxcr4* gene had interactions in the central nervous system (CNS) and was crucial for neurogenic processes, including neural migration and neural regeneration, particularly after brain injury [64,65,66]. Therefore, further investigation into the role of *cxcr4* in *C. semilaevis* is needed.

### 4.3. Heat Stress Upregulated Hormone-Related Genes in Females

Interestingly, significantly enriched categories, such as cortisol synthesis and secretion related genes, TGF-beta signaling pathway related genes, and the thyroid-stimulating hormone receptor (*tshr*) gene were up-regulated in both sexes under high temperatures. The TGF-beta signaling pathway plays a key role in cellular and tissue growth, development, and differentiation [67,68,69]. Currently, it is well established that higher temperatures activate the brain–pituitary–interrenal axis [14,70], increasing the plasma cortisol levels in several fish species [71,72,73]. Moreover, the involvement of thyroid hormones (THs) in response to high temperatures has been recently reported in medaka [14]. In this study, high temperatures activated thyroid signaling inducing female-to-male reversal and displayed high expression of the thyroid-stimulating hormone receptor (*tshr)* in *Oryzias latipes* embryos. Therefore, our results followed the patterns of the literature, and it might seem advisable to deeply study the effects of cortisol and THs in *C. semilaevis*.

The PPI analysis and heat map analysis revealed that numerous genes linked to the HSP family were present in females at first 1 h to 2 h, including *hsp70*, *hspa5*, *hspa41*, *hspb6*, and *hsp90*. The HSP family’s expression, including *hsp70* and *hsp90*, which are modulated under high temperatures has also been reported in various other species [74,75,76,77,78,79]. Moreover, it has been reported that *hsp70* and *hsp90* interact with the steroid hormone receptors to allow efficient hormone binding [80,81]. Our study showed that high temperatures up-regulated estrogen pathways and both *hsp70* and *hsp90* families in *C. semilaevis*. In this sense, our results may indicate that the *hsp70* and *hsp90* families played important roles in the interaction with the estrogen signaling pathway under high temperature.

RNAseq also showed that the estrogen signaling pathway involved in the regulation of female sexual differentiation and some genes involved in progesterone syntheses (*pgr*, *pgrmc1*, and *pgrmc2*) were up-regulated in female brains. In agreement with our results, other studies in pejerrey and tilapia brains also demonstrated that the estrogen pathway was modulated by temperature [25,82]. In addition, the gonadotropin-releasing hormone receptor (*gnrhr*) was also up-regulated in females under heat stress in the current study (Appendix A). This result was consistent with previous studies in salmonids, in which high temperatures altered gonadotrophin-releasing hormone (GnRH) secretion, gonadotrophin clearance, and gonadal steroidogenesis [23,24,26,83,84]. Nevertheless, a related study showed that higher GnRH expression in sea bass females’ brains could trigger gonadal development via the GnRH stimulation of FSH synthesis and release [85]. Moreover, it is known that FSH from the pituitary secretes estrogen to stimulate vitellogenesis and secretes progesterone to promote oocyte meiosis, follicular maturation, and ovulation [86]. In this sense, the upregulation of *pgr*, *pgrmc1*, and *pgrmc2* in our study could be associated with the high *gnrhr* expression in the brain. However, further studies are necessary to deeply understand this relationship.

### 4.4. Some Findings about DEGs in Male

We observed that both sexes exhibited enrichments in calcium signaling pathway and neuroactive ligand-receptor interactions in the KEGG enrichment analyses of cluster 3, which was consistent with the study in the brain of *Gymnocypris przewalskii* under high temperatures [87]. We focus on investigating this observation and hypothesize that there may be a relationship between neuroactive ligand–receptor interactions and the calcium signaling pathway. Xu D. [88] discovered that the cold-stress transcriptome of the yellow drum (*Nibea albiflora*) brain also exhibited neuroactive ligand–receptor interactions and the calcium pathway. A similar observation was made in the gonadal transcriptome of the heat-shocked *C. semilaevis* [89]. Additionally, Feng C. et al. showed that the brain of silver carp (*Hypophthalmichthys molitrix*) was rich in neuroactive ligand–receptor interactions and the calcium signaling pathway under hypoxic stress [90]. Moreover, in fish, both channel catfish (*Ictalurus punctatus*) [91] and grass carp (*C. idella*) [92] concurred with our findings, calmodulin related genes were also appeared in their findings. Finally, in males, our results showed that the heat stress treatment upregulated the transcription levels of important growth factors, including *igf2* and the receptors *igf2r* and *igf2bp2*, which is consistent with other studies [26,93,94,95,96,97]. All of the above evidence points to the need for further in-depth studies to establish the regulatory relationship between these pathways and genes.

## 5. Conclusions

Our study was the first to establish a brain transcriptome of male and female *C. semilaevis* exposed to early heat stress. This study identified several high-temperature response pathways and genes in both sexes, and some of them showed a sex-biased characteristic. This study offered valuable insights into the regulatory mechanisms and networks involved in early response to heat stress in fish and enhances our comprehension of gene expression effects in the brains of marine benthic fish. Future researches are needed to determine the function of these regulatory networks that respond to heat stress.

## Figures and Tables

**Figure 1 animals-14-00084-f001:**
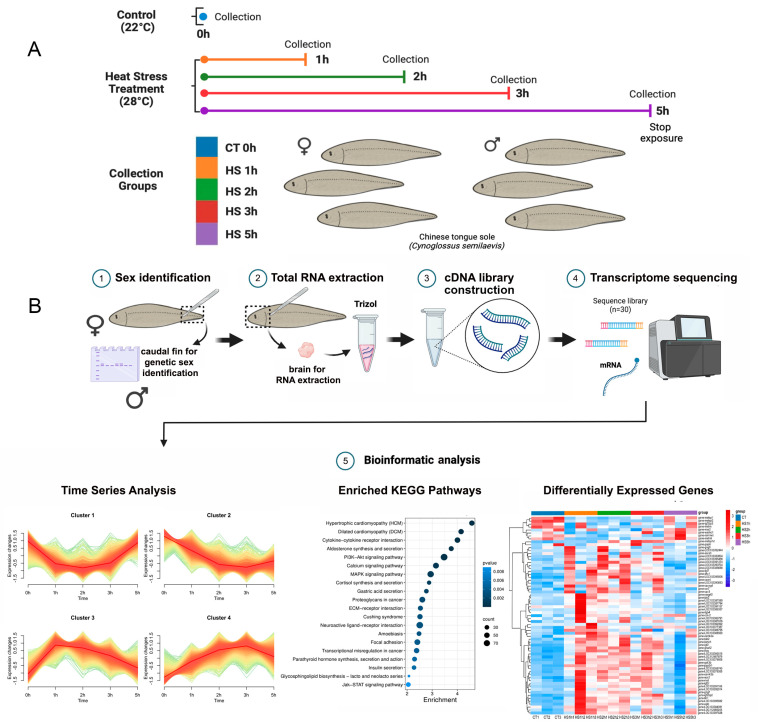
Profiling of high-temperature treatment experiment and RNA-seq analysis. (**A**) Schematic illustration of *C. semilaevis* subjected to both control (CT—22 °C) and heat stress (HS—28 °C) treatments. Samples from the control group were collected before treatment, while samples from the heat stress group were collected at 1, 2, 3, and 5 h post-treatment. (**B**) Sequential methodologies for transcriptome analysis: 1. Genomic DNA extraction from caudal fin samples for genotypic sex determination. 2. Total RNA extraction from brain samples using Trizol reagent. 3. cDNA library construction. 4. Transcriptome sequencing. 5. Bioinformatic analysis including time series analysis, enriched KEGG pathways analysis, and differentially expressed gene analysis were conducted subsequently. Figure created using BioRender.com.

**Figure 2 animals-14-00084-f002:**
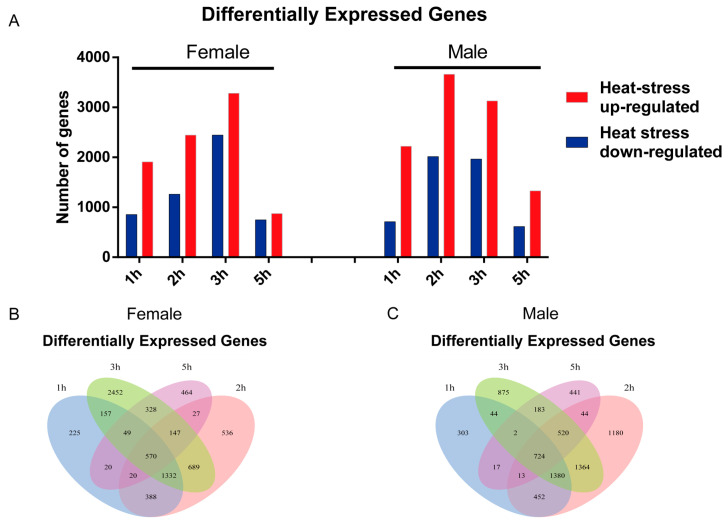
Differentially expressed gene analysis in females and males after heat shock. (**A**) Number of DEGs identified between control (CT) and heat stress (1 h, 2 h, 3 h, and 5 h) groups from females and males. Red color indicates up-regulated and blue color indicates down-regulated genes in HS vs. CT, respectively. (**B**) In females, distribution of DEGs depicted by a Venn diagram comparing the heat stress group (1 h, 2 h, 3 h, and 5 h) with the control group. (**C**) In males, the distribution of DEGs depicted by a Venn diagram comparing the heat stress group (1 h, 2 h, 3 h, and 5 h) with the control group.

**Figure 3 animals-14-00084-f003:**
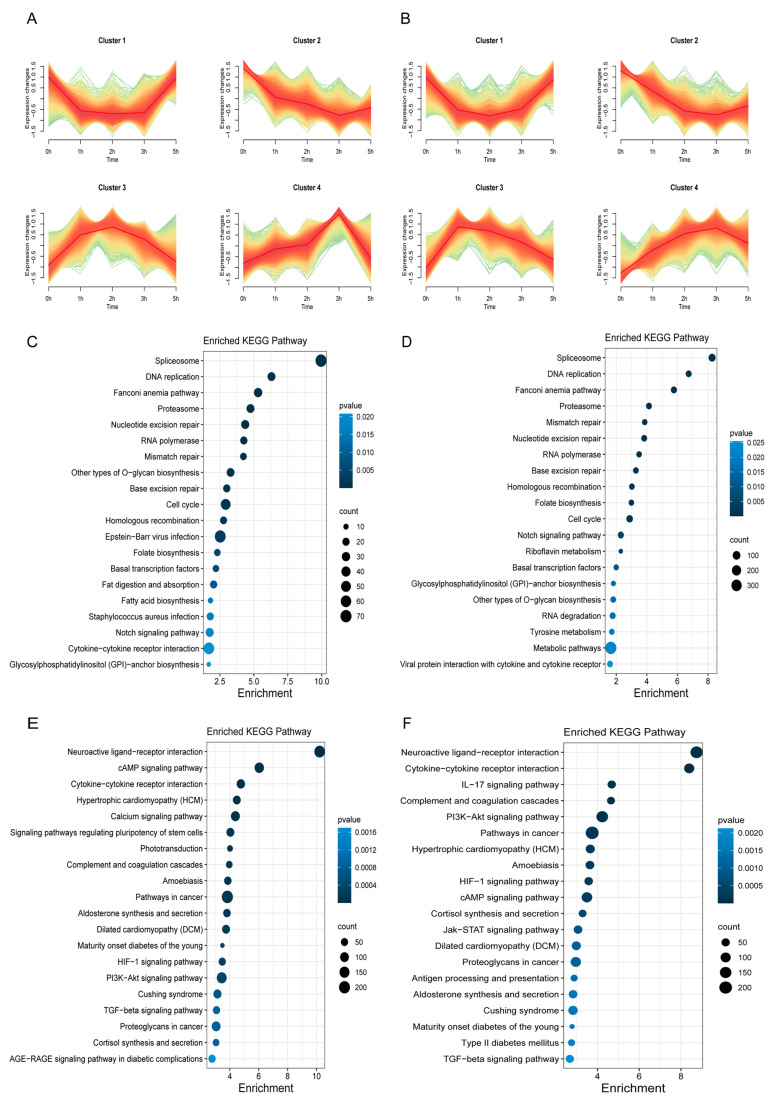
Profiling of transcriptome time-series DEGs and enriched KEGG analysis of clusters. (**A**) Female time-series clustering analysis. (**B**) Male time-series clustering analysis. The graph color transitions from green to red to indicate gene expression trends from discrete to clustered. (**C**) KEGG enrichment analysis of DEGs contained in Cluster 1 and Cluster 2 (U shape) of females. (**D**) KEGG enrichment analysis of DEGs contained in Cluster 1 and Cluster 2 (U shape) of males. (**E**) KEGG enrichment analysis of DEGs contained in Cluster 3 and Cluster 4 (bell shape) of females. (**F**) KEGG enrichment analysis of DEGs contained in Cluster 3 and Cluster 4 (bell shape) of males.

**Figure 4 animals-14-00084-f004:**
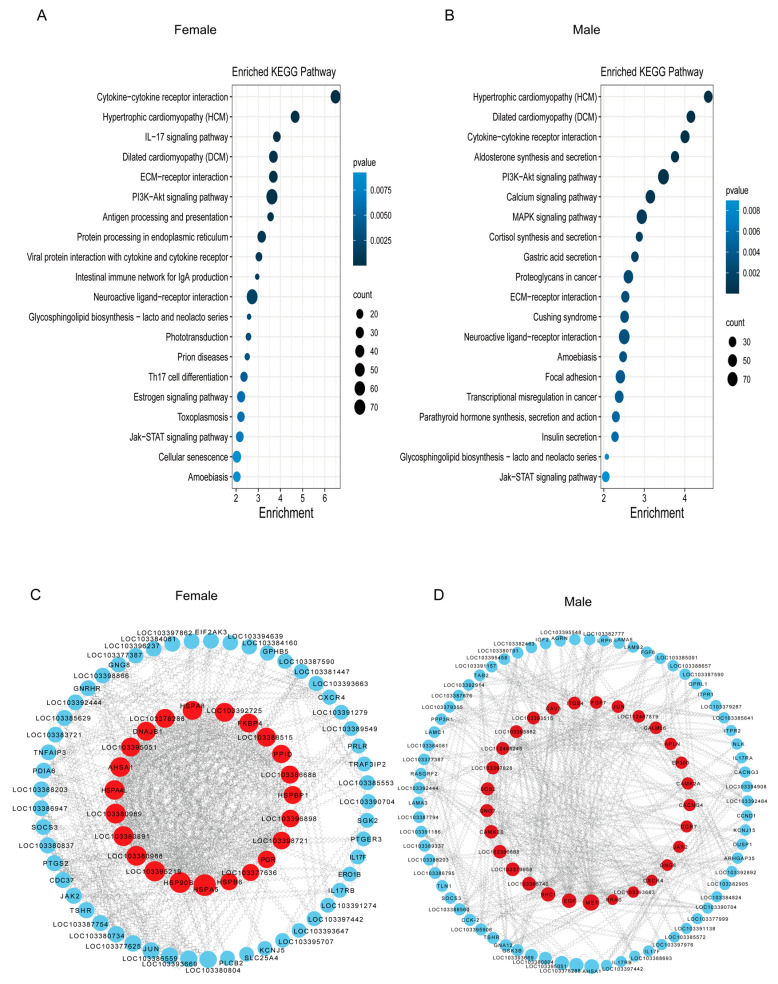
Analyses of cluster 3 in females and males. (**A**,**B**). KEGG analysis results for cluster 3 in both females and males, respectively. (**C**,**D**). Gene relationship maps in females and males, highlighting the selected genes, respectively. Red indicates more important key genes compared with blue.

**Figure 5 animals-14-00084-f005:**
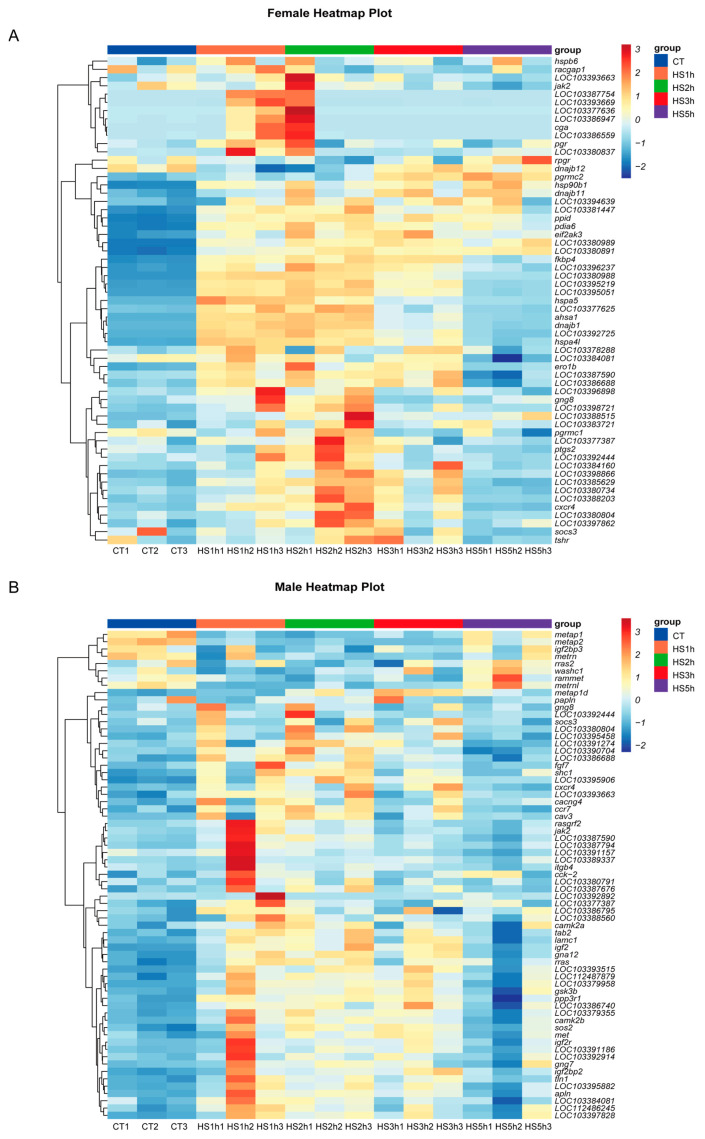
Heat map illustrating the relative expression of genes in females and males under heat stress. (**A**) Heat map of gene expression patterns in females derived from PPI analyses. (**B**) Heat map of gene expression patterns in males derived from PPI analyses. The rows represent the “expression levels of FPKM” and the columns represent treatment groups: control (CT1–3 at 22 °C) and heat stress (HS1h–HS5h at 28 °C). The higher expression values are displayed in red and lower expression values in blue.

## Data Availability

Data are contained within the article.

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
