# Peer review of "Transcriptome Analysis Revealed the Early Heat Stress Response in the Brain of Chinese Tongue Sole (Cynoglossus semilaevis)"

_animals, 2023, doi:10.3390/ani14010084_

Round 1

Reviewer 1 Report

Comments and Suggestions for Authors

In this article, Wang and colleagues conducted a comprehensive study to investigate the response of the Chinese tongue sole (Cynoglossus semilaevis) to temperature variations, with a particular focus on the transcriptomic profile of the entire brain. The study successfully identified specific genes and signaling pathways associated with the temperature response, shedding light on crucial mechanisms, including cortisol synthesis, interactions between neuroactive ligands and receptors, TGF-beta signaling, and JAK/STAT signaling.

This research makes significant contributions by illuminating the role of the brain in temperature adaptation, offering valuable insights for future exploration. It underscores the importance of understanding temperature-dependent mechanisms in aquatic organisms, particularly economically significant species such as the Chinese tongue sole.

However, the study does have a notable limitation in that it concentrates on the entire brain rather than specific brain regions. In recent years, advancements in fish neuroendocrinology have led to a more comprehensive understanding of specific brain areas and nuclei relevant to heat stress responses. Notably, the forebrain, especially the hypothalamus and telencephalon, have been linked to these responses. Thus, it prompts the question of why a more focused and specific study was not undertaken.

Additionally, several points in the manuscript require attention for improvement:

-        The authors used RNA with a quality indicator (RIN) greater than 7 for mRNA library construction. It is conventional to consider RNA of good quality when the RIN value exceeds 8. Clarification is needed regarding the choice of this slightly lower threshold.

-        Solely utilizing beta-actin for gene expression normalization is questionable without providing information on the stability of this reference gene. It is well-documented that beta-actin can fluctuate in some cases. Current standards recommend using multiple reference genes for accurate gene expression quantification.

-        On line 338, the term 'interleukin' is hyperlinked and displayed in blue. This may need correction based on journal formatting guidelines.

-        On line 468, there appears to be a red underscore under the word 'Finally.' Please verify whether this is accurate or not according to the journal's proofreading standards.

-        The enlarged views of Figures 4C, 4D, 4E, and 4F are not clear, and it is challenging to extract substantial information from them.

-        The study suggests that males may be more sensitive to acute heat stress than females, potentially resulting in differences in downstream gene regulation. It would be valuable to discuss these findings in the context of existing literature, exploring whether similar observations have been made in other species.

-        In order to enhance the article's utility, it is recommended to include indications of potential future research directions that merit further exploration.

Overall, this study presents a significant contribution to our understanding of temperature adaptation in Chinese tongue sole, but these suggestions and improvements can elevate the quality and impact of the manuscript.

Reviewer 2 Report

Comments and Suggestions for Authors

Temperature has profound effects on physiology of the ectothermic animals. As fish are ectothermic animals, temperature is one of the major abiotic environmental factors determining the main parameters of fish vital activity. In the research, the marine benthic fish Chinese tongue sole (Cynoglossus semilaevis) is used as a model to analyse the mechanisms governing the response to temperature remain unclear in teleost fish. However, there are some issues in the manuscript which need to be addressed by the authors.

1.      Line 74: Reference is needed.

2.      Line 109: Whether ethical approval was obtained for this study? If yes, please mention the “Ethical Statement”.

3.      Line 113: Please provide references for the time of the heat stress (HS) group exposed to 28℃.

4.      Line 173: Please elaborate on the steps.

5.      Line 174: Please provide qRT-PCR primers.

6.      Line 176; 345: RT-qPCR or qRT-PCR? Please provide the same word.

7.      Line 308: The image quality is too low, which makes it hard to view the details. The figures E and F would benefit from a higher resolution and the figures A, B, E, and F would benefit from focused highlights.

8.      Line 348: The results of qRT-PCR may be shifted to the result part.

9.      Line 22, 260: Please explain the function of neuroactive ligand-receptor interaction, TGF-22 Beta signaling pathway, and JAK/STAT signaling pathway. 

10.   Line 364: 1h of heat stress treatment is the response over a short period of time. Gene expression after stress treatment for a long period needs to be observed.

11.   Line 364–472: Add subheadings for easy understanding.

12.   Line 478: What insights do these results provide for our future research on what research?

13.   Line 605, 633, 635, 694, 696, 698, 703, 706, 708: Some references need to be formatted according to the journal's author guidence.

Reviewer 3 Report

Comments and Suggestions for Authors

The manuscript describes the effects of heat stress on gene expression in C. semilaevis brain while most of the research has been conducted in gonads for this fish. The heat stress was performed in a time series and the data have been accordingly analysed. The aim was to identify genes and pathways that are modulated by heat stress to lead future research. This work might contain some useful data for the readers. However, questions still need to be answered and amended before being published in "Animals". The specific points are shown as follows

Line 111: Please detail the salinity used for the acclimatisation

Line 112: Please rephrase.

Line 125: What does mean “conventional protocol”? If the authors have used a kit please provide its name.

Line 144: As the fold change could be either positive or negative the term “|fold change|” could be more appropriate than only “fold change”.

Line 149: How was the clustering of genes exhibiting analogous expression performed? Was it based on the data curation manually or using software?

Line 200: it is true that the CT group is separate from the HS groups in the PCA and that the HS1h and HS2h for each sex are the most different from the CT groups, however, the PCA must be reviewed as it is not very easy to read. Especially, in the PCA with both sexes, the colour code is not readable as it seems to be a colour gradient. Furthermore, there is no need to use one shape per replicate especially if no replicate was removed from the analysis.

Line 208: “…the control group and the 1h high-temperature…”  is the 1h missing?

Line 231: Concerning the time-series expression, it is very interesting, however, would it be more accurate to call the cluster “U or bell-shaped” according to the shape of the data more than up or down? Indeed, the HS5h condition is often closer to the CT group (as it is shown in the PCA and Figure 2A).

Line 247: Could you please provide more information about the way you choose the KEGG pathways which are the most enriched, i.e. based on the Rich Factor, the Q-value or a combination of both? Furthermore, it does not seem that the Q-value or Rich Factor are the same as in Tables S4 and S5 in Figure 3. For example, the “Cytokine-cytokine receptor interaction” pathway in females in Table S4 shows a Q-value of 0,0001069027 but the colour code in Figure 3C doesn’t seem to correspond. In Figure 3D for the same pathway, the colour code seems to correspond to Table S4 but not the Rich factor. Also, shouldn’t it be four tables instead of 2 (one for each merged cluster and both sexes)?

Line 286: Same comment as earlier on the KEGG pathways. For example, the authors write that the “Estrogen signaling pathway” is particularly enriched in females (lines 288-289) while this pathway shows a low Rich Factor and high Q-value in Figure 4A. Also, could the authors provide the tables corresponding to the KEGG enrichment?

Line 362-363: Are there other studies that pointed out sex-specific effects in response to acute HS in this species or others that could support the authors finding concerning males being more sensitive than females?

Figure 1: The caption indicates that the control group were collected 1h after treatment however in the figure the collection is set to 0h. Please correct the typo.

Figure 3: The figure is a bit small and shows low resolution in the PDF which makes it difficult to read even when zooming. The colour code is missing for panels A and B.

Figure 4 A and B are too small and have low image resolution.

Figure 4 CDEF are too small and have low image resolution. Furthermore, they are very difficult to read therefore it seems not relevant to add them to the manuscript, they could be moved to the supplementary materials. Authors could however make a new figure based on the gene degree values which is more developed in the second paragraph of the 3.4 section.

Figure 5: replace “brain” by “gene”

Round 2

Reviewer 3 Report

Comments and Suggestions for Authors

The authors kindly responded to the questions in the review and have made the modifications in the manuscript. Therefore, the manuscript could be accepted in this reviewed form.